# Molecular Mechanisms and Emerging Therapeutics for Osteoporosis

**DOI:** 10.3390/ijms21207623

**Published:** 2020-10-15

**Authors:** Ji-Yoon Noh, Young Yang, Haiyoung Jung

**Affiliations:** 1Immunotherapy Research Center, Korea Research Institute of Bioscience and Biotechnology (KRIBB), 125 Gwahak-ro, Yuseong-gu, Daejeon 34141, Korea; nohj16@kribb.re.kr; 2Research Institute of Women’s Health, Sookmyung Women’s University, 100 Cheongparo47gil, Yongsan-Gu, Seoul 04310, Korea; 3Department of Functional Genomics, Korea University of Science and Technology (UST), 113 Gwahak-ro, Yuseong-gu, Daejeon 34113, Korea

**Keywords:** osteoporosis, fracture, medication, molecular mechanism, novel approach

## Abstract

Osteoporosis is the most common chronic metabolic bone disease. It has been estimated that more than 10 million people in the United States and 200 million men and women worldwide have osteoporosis. Given that the aging population is rapidly increasing in many countries, osteoporosis could become a global challenge with an impact on the quality of life of the affected individuals. Osteoporosis can be defined as a condition characterized by low bone density and increased risk of fractures due to the deterioration of the bone architecture. Thus, the major goal of treatment is to reduce the risk for fractures. There are several treatment options, mostly medications that can control disease progression in risk groups, such as postmenopausal women and elderly men. Recent studies on the basic molecular mechanisms and clinical implications of osteoporosis have identified novel therapeutic targets. Emerging therapies targeting novel disease mechanisms could provide powerful approaches for osteoporosis management in the future. Here, we review the etiology of osteoporosis and the molecular mechanism of bone remodeling, present current pharmacological options, and discuss emerging therapies targeting novel mechanisms, investigational treatments, and new promising therapeutic approaches.

## 1. Introduction

Osteoporosis occurs due to an imbalance between bone resorption and bone formation. As a result, bone breakdown exceeds bone formation. In 1993, the World Health Organization (WHO) defined osteoporosis as a “progressive systemic skeletal disease characterized by low bone mass and microarchitectural deterioration of bone tissue, with a consequent increase in bone fragility and susceptibility to fracture” [1,2,3]. Osteoporosis is a highly prevalent disorder estimated to affect 200 million women and men worldwide, predominantly those over the age of 60 years. [4]. Osteoporotic fracture is a major health concern that significantly impacts the quality of life of the affected individuals. According to the International Osteoporosis Foundation, worldwide, one in three women and one in five men over the age of 50 years will experience osteoporotic fractures in their lifetime. On the other hand, more than 8.9 million fractures are caused by osteoporosis annually, which means that an osteoporotic fracture occurs every three seconds. Approximately 33% of patients experience a hip fracture and in the year following the fracture, up to 20% die, mainly due to preexisting conditions [5].

Given that life expectancy is increasing globally, osteoporosis will affect the quality of life of individuals and impose an economic burden in most countries. Therefore, osteoporosis should be properly managed using effective approaches, and this can be achieved by understanding the mechanisms underlying the pathogenesis of this disease. To date, bisphosphonates (BPs), which inhibit bone resorption, are one of the most common medications for the treatment of osteoporosis. However, due to the adverse effects and limited efficacy of the currently used medications, pharmaceutical companies have been trying to move beyond BPs. This review will discuss the etiology of osteoporosis and the molecular mechanism of bone modeling, current pharmacological options, emerging therapies targeting novel mechanisms, investigational treatments, and new promising therapeutic approaches.

## 2. Nature of Osteoporosis and Its Etiology

Osteoporosis is defined as a bone mineral density (BMD) T score of −2.5 or less, according to the diagnostic criteria produced by WHO using standard deviation scores of BMD related to peak bone mass in healthy young women. BMD T scores between −1 and −2.5 are considered osteopenia and low bone mass [6]. However, BMD is only one of the risk factors for fracture, and the majority of fragility fractures occur in individuals with BMD T values above the −2.5 threshold, suggesting that BMD is a limited indicator of osteoporosis in the clinic [7,8]. Deterioration of bone architecture increases the risk of fracture. Osteoporotic fractures are the primary cause of bone-associated morbidities and lead to a 2–8-fold increased risk of mortality [9,10]. For instance, an 8–36% increased mortality risk was found within one year after a hip fracture [11]. Fractures in the population affected by osteoporosis determine the quality of life, as they cause pain, impaired mobility, substantially reduce pulmonary function [12], affect the risk of infection, lead to changes in body image, psychosocial distress, social isolation, loss of independence, and eventually, determine life expectancy [13,14]. Thus, drug discovery efforts aim to reduce the fracture rate and increase BMD.

Bones are continuously resorbed by osteoclasts and replaced with new bones formed by osteoblasts [15]. This bone remodeling activity can provide mechanical strength to the bone and repair it. It is known that, normally, bone resorption lasts around four to six weeks whereas bone formation lasts approximately four to six months. The balance between bone resorption and bone formation and the regulation of these processes are critical for maintaining bone density and mineral homeostasis under healthy conditions. Osteoporosis occurs as a result of an imbalance in the remodeling process in which bone resorption exceeds bone formation. There are several conditions that can cause an imbalance in the remodeling process leading to osteoporosis. Whereas the postmenopausal period and old age are the major causes of osteoporosis, other risk factors including medications, endocrine disorders, immobilization, inflammatory arthropathy, hematopoietic disorders, nutrition disorders, can also be involved. Osteoporosis could certainly be categorized according to its causes and each cause could be treated separately. It is now more important than ever before to understand the underlying pathophysiology and provide a proper diagnosis and management of symptoms. The etiology of osteoporosis is further described below (Figure 1).

### 2.1. Postmenopause and Age

Hormones, such as estrogen, testosterone, and parathyroid hormone (PTH), play a significant role in bone remodeling by inhibiting bone breakdown and promoting bone formation. The peak bone mass is reached at the age of 25–30 years in women. The reduced production of estrogen in postmenopausal women induces substantial bone loss. Whereas bone loss is accelerated during the perimenopausal period, the rate of bone loss decreases several years after menopause. On the other hand, a gradual decrease in BMD is observed in men. With aging, sex-hormone-binding globulin is considered to inactivate testosterone and estrogen and may be involved in the progressive decline of BMD [16]. Thus, due to age-related hormonal changes, men and women have equal rates of bone loss by the age of 60 years and are at risk of developing osteoporosis [17,18]. Negative bone remodeling in elderly women is associated with both cancellous and cortical bone, the disruption of bone microarchitecture and bone loss. Trabecular thinning is observed in cancellous bone, whereas reduced cortical thickness and increased cortical porosity are seen in cortical bone [19,20]. On the other hand, bone loss is predominantly related to reduced bone formation and low bone turnover in men.

### 2.2. Secondary Osteoporosis

Glucocorticoids, used to suppress various allergic, inflammatory, and autoimmune diseases, and to prevent graft-versus-host disease after transplantation, are the most common cause of drug-induced osteoporosis. The long-term use of glucocorticoids can result in complications such as glucocorticoid-induced osteoporosis (GIO) [21,22]. Glucocorticoids enhance the differentiation and maturation of osteoclasts [23], while inhibiting osteoblastogenesis by promoting osteoblast and osteocyte apoptosis [24], resulting in increased bone resorption and decreased bone formation [25,26]. Glucocorticoids also suppress insulin-like growth factor 1 (IGF1), which promotes bone formation by stimulating type I collagen synthesis, leading to collagen degradation and osteoblast apoptosis. In GIO, a rapid decline of BMD has been observed within three to six months of beginning of glucocorticoid therapy [27]. Chronic low-dose glucocorticoid treatment in men receiving androgen-deprivation therapy for prostate cancer has been associated with an increased risk of osteoporosis and the occurrence of more fractures, compared to control individuals [28]. Furthermore, patients with Cushing’s syndrome may also suffer from accelerated bone loss due to excess glucocorticoid production [29].

### 2.3. Heritable Factor on Osteoporotic Fractures

Efforts have been made to identify the mechanisms underlying osteoporosis using omics technologies [30,31]. Although the current functional genomics and other omics studies of osteoporosis have limitations, associated especially with the lack of healthy human control samples, it is important to continue investigations in this field as the currently used diagnostic method, BMD, cannot predict who will experience an osteoporotic fracture. It is expected that a multi-omics analysis will reveal the precise mechanism underlying osteoporotic fractures and identify high-risk patients more easily in the future [32,33].

For instance, it has been demonstrated that a missense variant (Gln63Arg) of *CNR2*, which was strongly associated with BMD in a population of postmenopausal osteoporosis patients, affects cannabinoid receptor type 2 (CB2) expression and activity [34]. Thus, a decrease in the expression or efficacy of CB2 signaling was suggested to affect lower bone density and even osteoporosis.

## 3. Molecular Mechanism of Bone Remodeling and Osteoporosis

### 3.1. Role of Bone Cells in Bone Loss

Osteoclasts, osteoblasts, and osteocytes are bone cells that are directly involved in bone remodeling [15]. Bone formation, by mesenchymal stem cell (MSC)-derived osteoblasts and bone resorption by tissue-specific macrophage polykaryons-derived osteoclasts are coordinated to maintain bone mineral homeostasis and strength. Bone loss can occur due to the malfunction of each cell type.

#### 3.1.1. Osteoclasts

Osteoclasts are large multinucleated cells that adhere to the bone and resorb it, resulting in the formation of holes and the release of calcium into the blood. These cells are differentiated from hematopoietic stem cells (HSCs) upon stimulation by monocyte/macrophage colony-stimulating factor (M-CSF) and activation of receptor activator of nuclear factor kappaB (RANK) with its ligand (RANKL) [35,36,37,38,39]. Other factors, including inflammatory cytokines, such as interleukin 1 (IL-1), IL-6, and tumor necrosis factor-α (TNF-α), and αVβ3 integrin, can also regulate osteoclast differentiation and function [40,41,42,43]. Once an interaction between the RANK receptor and its ligand occurs, molecules such as TNF receptor-associated factor 6 (TRAF6) are recruited, leading to the activation of the MAPK cascade, NF-κB, AKT/PKB, JNK, and ERK downstream signaling pathways, and the expression of genes involved in osteoclastogenesis.

The PI3K-AKT pathway plays a pivotal role in osteoclast differentiation and activation [44,45,46,47]. Recently, the Rho GTPase family member RhoA was reported to suppress AKT activity and negatively control osteoclastogenesis [48], mediated by guanine nucleotide-binding protein subunit α13 (Gα13), upstream of RhoA during cell cytoskeleton organization. The absence of Gα13 favors osteoclast formation and enlarges osteoclast size, resulting in bone degradation in mice. In line with this, constitutively active Gα13 (Ga13CA) prohibited the formation of multinucleated osteoclasts. In fact, Ga13CA over-expressing monocytes were TRAP-positive, indicating that Ga13CA modulated the late stage of osteoclast maturation rather than earlier stages, and could target bone loss without affecting normal bone remodeling [48]. Moreover, genome-wide studies have identified that single nucleotide polymorphisms (SNP) in *RHOA* and *ARHGEF3* (RhoGEF3) were significantly associated with decreased BMD in postmenopausal women [49,50,51]. The authors suggested that the activity of Gα13 and RhoA, mainly mediated by the downregulation of osteoclast formation and bone resorption, is important in osteoporosis [52].

Osteoclasts induce bone resorption, a process of mineral dissolution and bone degradation, through secreting proteolytic enzymes and hydrochloric acid [35,40,53,54]. The important proteolytic enzymes released from osteoclasts are lysosomal enzymes (e.g., cathepsin K) and matrix metallopeptidase 9 (MMP-9) [55,56,57]. This can occur in response to parathyroid hormone (PTH) and calcitonin stimulation. PTH-activated osteoclasts can release minerals back into the bloodstream, as a part of the mechanism of calcium homeostasis [15]. PTH can also indirectly increase osteoblast proliferation.

#### 3.1.2. Osteoblast

Osteoblast-induced development of new bone begins in the embryo approximately six weeks after fertilization. Bone formation can be divided into two types of ossification: intramembranous and endochondral [58]. The former involves in a crucial process occurring during the natural healing of fractures and the formation of the flat bones of the clavicles and skull. Endochondral ossification is a process related to the formation of long bones, cartilage replacement, and healing of bone fractures [59,60]. During intramembranous bone formation, MSCs proliferate and differentiate into osteoblasts, which produce bone by synthesizing extracellular matrix proteins, such as type I collagen, the most abundant one. Once deposited, the extracellular matrix is subsequently mineralized through the accumulation of calcium phosphate as hydroxyapatite (Ca_10_(PO_4_)_6_(OH)_2_) [61].

Signaling molecules with crucial roles in osteoblast turnover are Runt-related transcription factor 2 (Runx2), osterix (Osx), β-catenin, activating transcription factor 4 (Atf4), and activator protein 1 (AP-1) family [62,63,64]. Runx2 is a key transcription factor involved in osteoblast differentiation. The level of Runx2 is increased by stimulation with bone morphogenetic proteins (BMPs) and Wnt (particularly, Wnt3a and Wnt10b), mediated through the activation of the Frizzled and lipoprotein receptor-related protein (LRP)-5/6 receptors [65], resulting in osteoblastogenesis, which promotes bone formation. Similarly, fibroblast growth factors (FGFs), transforming growth factor-β1 (TGF-β1), IGF-1, Notch, and PTH have also been shown to promote bone formation [66,67,68,69]. For example, during skeletal remodeling, TGF-β1 is released from the bone matrix and recruits MSCs, which further generate osteoblasts [70].

Osteoblasts, in addition to forming bones by synthesizing extracellular matrix, regulate bone mass by modulating osteoclasts, positively or negatively. RANKL is a homotrimeric transmembrane protein that is expressed by osteocytes, macrophages, osteoblasts, bone marrow stem cells, and activated T lymphocytes [71,72]. The prominent role of RANKL expression on osteoblast surface is to promote the differentiation of osteoclasts through cell-to-cell-dependent contact activation. RANKL also inhibits osteoclast apoptosis. Importantly, genetic mutations in the human RANKL gene and RANKL knockout mice were associated with osteoclast deficiency and severe osteosclerosis, suggesting that osteoblasts play a critical role in bone remodeling [73,74].

Osteoprotegerin (OPG) is a soluble secreted protein lacking a transmembrane domain and a cytoplasmic domain that is principally expressed by osteoblasts and bone marrow stromal cells. OPG is a decoy receptor of RANKL that competitively binds to the trimer RANKL, preventing RANKL-induced osteoclast maturation and promoting osteoclast apoptosis [15,75]. Interestingly, OPG can bind RANKL with an affinity approximately 500 times higher than that of RANK [76]. Thus, the OPG/RANKL ratio is important for maintaining bone density and bone strength, and downregulation of OPG might trigger osteoporosis and bone loss, associated with pathological bone disorders such as rheumatoid arthritis and Paget’s disease [77].

#### 3.1.3. Osteocyte

Osteocytes have gained attention for their central role in bone remodeling. As one of the major cellular components of bone tissue, osteocytes are completely embedded in the bone matrix and comprise more than 90% of all bone cells [78]. Osteocytes originate from MSCs-derived osteoblasts, which can orchestrate bone formation by secreting stimulators of the WNT signaling pathway, such as nitric oxide and ATP as well as inhibitors such as sclerostin and Dickkopf-related protein 1 (DKK1). Osteoblast function is impaired by sclerostin and DKK1, mediated through inhibition of WNT/β-catenin signaling and decrease in Runx2 expression [79,80]. Osteocytes can modulate osteoclast activation or inhibition by expressing RANKL and M-CSF, or nitric oxide and OPG, respectively [81,82].

#### 3.1.4. Adipocyte

Bone marrow adipogenesis plays a critical role in bone loss associated with aging as well as diseases such as diabetes mellitus. The differentiation of adipocytes derived from MSC in bone marrow competes with osteoblastogenesis by its nature of lineage allocation while the mature adipocytes express RANKL and promote osteoclastogenesis [83,84]. Notably, among the adipogenic transcription factors, including CCAAT/enhancer binding protein alpha (C/EBPα), C/EBPβ, and peroxisome proliferator-associated receptor gamma (PPARγ), PPARγ is considered as a master factor for osteoclast differentiation from HSCs [85]. PPARγ downregulates Wnt/β-catenin signaling pathway in both MSC and HSC, resulting in inhibition of osteoblastogenesis and activation of osteoclast function, respectively [86]. In osteoclast, PPARγ-mediated expression of peroxisome proliferator-activated receptor-gamma coactivator-1 beta (PGC1β) promotes mitochondria biogenesis, which in turn increases bone resorption [87].

In addition to the regulation by transcription factors in bone cells and adipocytes, paracrine effects by adipokines such as chemerin, resistin, visfatin, leptin, adiponectin, and omentin-1 secreted from adipocytes have been shown to involve in bone remodeling. Interestingly, a cognate receptor for chemerin, chemokine-like receptor 1 (CMKLR1) was identified in MSCs, and chemerin/CMKLR1 signaling suppresses Wnt/β-catenin and Notch signaling pathways, resulting in inhibition of osteoblastogenesis [86]. Furthermore, chemerin/CMKLR1 signaling induces expression of NFATc1 in HSCs, a key transcription factor for osteoclast differentiation, demonstrating that chemerin may potentiate bone loss in disease states [88]. Other adipokines such as visfatin and resistin represented similar activity, although the role of visfatin in bone remodeling is controversial [83]. Collectively, bone formation is regulated by the cell-to-cell contact involving RANK-RANKL in normal condition, however, adipocytes as well as adipokines can play critical roles in bone-loss-associated diseases.

### 3.2. Other Factors for Osteoporosis

GIO is the most common form of secondary osteoporosis [89]. Several studies have demonstrated that a chronic glucocorticoid therapy is strongly associated with a low BMD and high susceptibility to fractures. Prednisolone treatment induces apoptosis of osteoblasts and osteocytes, leading to reduction of bone formation [90].

Iron is also an important risk factor for osteoporosis [91]. Iron overload is generally a consequence of chronic blood transfusions that are necessary in disorders such as beta thalassemia major, hereditary hemochromatosis, and sickle cell anemia. Iron overload in mice results in increased bone resorption and oxidative stress, leading to changes in bone microarchitecture and bone loss. Iron can directly inhibit osteogenic commitment and bone marrow stromal cell differentiation [92]; however, the underlying mechanism remains unknown.

## 4. Therapeutic Approach and Novel Strategies

Numerous medications and therapeutic options have been established for the treatment of osteoporosis [93]. As osteoporosis occurs as a result of an imbalance between bone resorption and bone formation, the pharmacological options for its management are anti-resorptive and anabolic agents. To assess drug efficacy, non-invasive methods such as dual-energy vertebral assessment program (DXA) or micro-computed tomographic (microCT) can be used. The principle goal of pharmacological therapy is to reduce the risk of osteoporotic fractures.

### 4.1. Anti-Resorptive Agents

#### 4.1.1. Bisphosphonates (BPs)

BPs come in close contact with osteoclasts and reduce bone resorption by inducing osteoclast apoptosis. BPs are stable analogs of inorganic pyrophosphate and have a core structure of P-C-P bonds, which are responsible for the strong binding affinity toward hydroxyapatite, the major mineral component of bone [94]. This binding to bone minerals enables BPs to be taken up by osteoclasts and inhibit their activity. BPs have been used for the treatment of osteoporosis since the 1990s. They are available in inexpensive generic form, in oral and intravenous formulations, and remain the first-line medications for the treatment of osteoporosis [95]. Alendronate, risedronate, and ibandronate are available as oral tablets, while zoledronic acid and ibandronate are used intravenously. BPs are approved for use in GIO patients who are at increased risk of fracture [96,97,98]. The major adverse event is the risk of atypical femoral fractures and osteonecrosis of the jaw. Gastrointestinal and renal complications have also been reported [99,100], and long-term use of BPs is also associated with a risk for osteomalacia [101]. BPs can be prescribed for less than five years and are supplemented with calcium.

#### 4.1.2. Denosumab

Denosumab (Prolia) is a human IgG2 monoclonal antibody against RANKL that inhibits osteoclast formation, function, and survival [102,103,104]. The half-life of denosumab is approximately 26 days and it does not appear to form neutralizing antibodies [105]. Denosumab is FDA-approved for the treatment of postmenopausal osteoporosis with a high risk for fracture as well as for bone loss in men with prostate cancer receiving androgen deprivation therapy. It has also been approved for women with breast cancer, who are at risk for osteoporotic fracture. A 60 mg dose is applied subcutaneously every six months and can be supplemented with oral calcium and vitamin D. In GIO, denosumab was shown to be more effective than risedronate in increasing BMD [106]. The adverse effects of denosumab are related to the fact that RANKL is also abundantly expressed by dendritic cells and activated T lymphocytes, and its antagonistic effect could affect the immune system [75]. In the previous study, it was reported that the denosumab treatment group showed skin eczema (3%) and cellulitis (0.3%) compared to the control group [107].

#### 4.1.3. Selective Estrogen Receptor Modulators (SERMs)

Estrogen has been shown to directly regulate the survival of mature osteoclasts via the Fas/FasL system [108]. Consistently, selective ablation of estrogen receptor alpha in the osteoclasts of women could lead to an osteoporotic bone-like phenotype. SERM or estrogen interacts with the RANKL/RANK/OPG system and decreases bone resorption [109]. Raloxifene, representing dual agonistic and antagonistic properties in estrogenic pathways (estrogen agnonist/antagonist, EAA), is a first-line therapy for patients with a high risk for spine fracture. It can also be prescribed to patients who have been treated with BPs. A combination of conjugated estrogens and bazedoxifene was approved by FDA for use in postmenopausal women for the prevention of osteoporosis and the treatment of moderate-to-severe vasomotor systems [12]. However, raloxifene is associated with a small increase in the risk of venous thromboembolism and stroke. On the other hand, as the overall health risks exceed the benefits, hormonal replacement therapy, such as estrogen-progestin treatment, is no longer recommended as first line therapy for the prevention of osteoporosis in postmenopausal women [110,111].

#### 4.1.4. Calcitonin

Calcitonin is a synthetic polypeptide hormone with the natural properties of calcitonin, which is found in mammals, birds, and fish. Calcitonin receptors are expressed on osteoclasts and osteoblasts. Calcitonin prevents osteoclast precursors from maturing and regulates osteoclast function [112,113,114]. This peptide hormone binds to receptors mainly located on the surface of osteoclasts, resulting in reduction of bone resorption activity. Furthermore, calcitonin has an analgesic effect [115]. Thus, it may be a preferred treatment for patients with acute osteoporotic fractures. Calcitonin gained FDA approval for the treatment of osteoporosis in postmenopausal women who have had osteoporosis for more than five years and could not receive alternative treatments. However, the calcitonin-salmon nasal spray has adverse effects, including rhinitis, nasal irritation, back pain, nosebleed, and headache [116].

### 4.2. Emerging Therapies and Investigational Agents Targeting Bone Resorption

#### 4.2.1. Cathepsin K Inhibitors

Cathepsin K, the primary enzyme released from osteoclasts, digests collagen in bones. It is a more desirable anti-bone resorption target as it prevents osteoclast activity by inhibiting the late differentiation of osteoclasts without affecting normal bone remodeling [117]. The advantage of targeting cathepsin K rather than osteoclastogenesis is to allow continued signals to osteoblasts and consequent bone formation. Odanacatib is a selective cathepsin K inhibitor; unfortunately, Merck discontinued the development of odanacatib due to an increased risk of stroke [118].

#### 4.2.2. Lasofoxifene

Lasofoxifene is a third-generation SERM. It is approved for osteoporosis treatment in Europe, but its approval is pending in the United States [119]. In a clinical study, the group treated with lasofoxifene at a dose of 0.5 mg per day demonstrated a 42% risk reduction for vertebral fractures and a 24% risk reduction for nonvertebral fractures. It has also been found that lasofoxifene treatment was associated with a decrease in breast cancer, coronary heart disease, and stroke occurrence [120]. Recently, the FDA granted a fast track designation to lasofoxifene for the treatment of women with estrogen receptor-positive, HER2-negative metastatic breast cancer.

### 4.3. Anabolic Agents

#### 4.3.1. PTH and Parathyroid Hormone-Related Protein (PTHrP) Analogues

PTH and PTHrP can increase the number and activity of osteoblasts by stimulating osteoblast differentiation, as a consequence of increased bone formation. PTH is secreted by the parathyroid gland to adjust homeostasis of serum calcium and phosphate mainly in response to low blood calcium levels. Binding of PTH to osteoblasts induces RANKL expression, which increases osteoclast differentiation and function, and results in calcium release by bone resorption. PTH can also reverse the glucocorticoids-induced IGF-1 suppression in GIO.

Teriparatide, the first anabolic treatment approved for osteoporosis, is a recombinant human PTH (1–34) analogue. It is well known that continuous PTH dosing results in a catabolic effect, and conversely, intermittent intake promotes an anabolic effect on bone [121]. Stimulation of osteoblastic activity was shown by intermittent administration of teriparatide at small doses [122], mainly mediated through expression of interleukin-11, suppression of DKK-1, and activation of Wnt signaling [123]. Although therapies with teriparatide showed great improvement on BMD, it is not clear whether teriparatide could prevent fracture efficiently [124,125]. In GIO, however, teriparatide treatment was related to significantly fewer new vertebral fractures compared to alendronate treatment at 18 and 36 months [126]. However, due to its risk for osteosarcoma, the usage of teriparatide is restricted to those at a very high risk for fracture.

Abaloparatide (PTHrP1-34), the second recombinant human PTH analog, received FDA approval in 2017. It is expected to induce a stronger anabolic effect than teriparatide [127]. In a phase three clinical trial, abaloparatide reduced the incidence of new vertebral fracture by 86% and nonvertebral fracture by 43% over an 18-month period [125]. Treatment with this drug is limited to two years. Furthermore, the use of abaloparatide is more cost effective than that of teriparatide.

#### 4.3.2. Strontium Ranelate

Strontium has the dual effects of preventing bone resorption and promoting bone formation [128]. It stimulates the differentiation of pre-osteoblasts into osteoblasts, and activates osteoblasts to release OPG, which can interfere with osteoclast differentiation by acting as a decoy receptor for RANKL. Thus, it increases bone formation and decreases bone resorption; however, due to the increased risk of heart problems, the use of strontium is restricted to severe osteoporosis where other treatments are not available.

### 4.4. Emerging Therapies and Investigational Agents for Targeting Bone Formation

#### 4.4.1. Anti-Sclerostin Antibodies

Sclerostin is an osteocyte/osteoclast-secreted protein that interferes with osteoblast differentiation, proliferation, and activity. It competitively binds to LRP-5/6 on osteoblasts and inhibits the Wnt/β-catenin pathway, thereby preventing osteoblast differentiation [129,130,131,132]. An anti-sclerostin antibody, romosozumab, showed a greater increase in BMD than alendronate and teriparatide in phase three clinical trials. Furthermore, it showed a 73% lower risk for new vertebral fracture at 12 months compared with placebo. In July 2017, the FDA rejected the approval of rosomozumab for osteoporosis treatment due to a higher rate of serious cardiovascular events compared with alendronate. In April 2019, romosozumab (Evenity, Amgen/UCB), a humanized monoclonal antibody, finally received FDA approval. It comes with the drug’s label noting an increased risk of myocardial infarction, stroke and cardiovascular death in clinical trials [133]. Other anti-sclerostin monoclonal antibodies, such as blosozumab and BPS804, are in the process of drug development [134].

#### 4.4.2. Donepezil

Donepezil is a medication that has been widely used in the treatment of Alzheimer’s disease and other dementias since the mid-1990s [135]. It is a reversible acetylcholinesterase (AChE) inhibitor. Notably, acetylcholine receptors (AChR) are expressed in bone cells, of which stimulation by inhibiting the action of AChE and increasing the amount of Ach can have anabolic effects in bone. In line with it, AChE is highly expressed on bone cells, especially during osteoblastogenesis [136], suggesting that AChE can be a therapeutic target for treating osteoporosis. Interestingly, donepezil showed beneficial effects on bone turnover, associated with reduction of hip fracture risk in Alzheimer’s disease patients [137,138].

#### 4.4.3. Suppression of Bone Marrow Adipogenesis and/or Adipokines (Chemerin)

Approaches to suppress bone marrow adipogenesis and/or chemerin may have therapeutic value for treating osteoporosis due to the advantages resulting from inhibition of bone resorption and enhanced bone formation [86]. In this context, PPARγ is a good target because of its dual role in MSC-derived adipogenesis as well as HSC-derived osteoclastogenesis. PPARγ can be directly targeted by its antagonist, GW9962 [139], or by regulation of upstream signaling molecules, for instance, lipid mediators. Recently, it has been reported that upregulation of sphingosine-1-phosphate (S1P) by inhibition of S1P lyase showed therapeutic benefit for treating osteoporosis, mainly mediated through PPARγ suppression and enhanced bone formation [140]. Similarly, the level of glucosylceramide (GlcCer) is also important for PPARγ activity, and inhibition of GlcCer synthase has been shown to suppress adipogenesis [141].

Therapeutic effect on osteoporosis can be also achieved by targeting adipokines such as chemerin and visfatin [86]. Since chemerin/CMKLR1 signaling inhibits Wnt/β-catenin and Notch signaling pathway, bone-anabolic activity can be reversed by suppression of chemerin. Interestingly, this suppression of adipogenesis was found to be restricted to the bone marrow adipose tissue. Furthermore, osteoporotic patients treated with bone anabolic agents (e.g., teriparatide) have shown less adipose tissue in bone marrow, indicating that osteoporotic therapies targeting the osteoblast/adipocyte/osteoclast complex are promising [142].

### 4.5. Non-Pharmacological Fracture Prevention

#### 4.5.1. Calcium

Calcium intake is the best option only in patients whose osteoporosis pathology is directly related to calcium shortage or patients with secondary hyperparathyroidism. Administration of calcium (800–1200 mg daily) will suppress PTH release and eventually decrease bone resorption and bone turnover. However, excessive calcium intake (more than 1500 mg total daily) is not beneficial, will be excreted, and it might be associated with an increased risk of renal stones [143,144].

#### 4.5.2. Vitamin D

Vitamin D modulates calcium metabolism, including intestinal absorption, renal excretion, and bone resorption. All patients receiving glucocorticoid therapy should improve nutrition to decrease fracture risk, and it can be partially achieved by adequate calcium and vitamin D status (serum level of 25-hydroxyvitamin D, > 20 ng/mL; 50 nmol/L). In order to correct the deficiencies, supplements can be used at a dose of 600–800 IU vitamin D daily. Several reports have shown that active vitamin D has positive effects in increasing BMD and preventing vertebral fractures [145,146]. On the contrary, intermittent high doses of vitamin D (60,000 IU monthly or 500,000 IU annually) have been associated with an increased risk of falls and fractures. Thus, the recommended daily dose should not exceed 4000 IU of vitamin D in normal status [147,148].

#### 4.5.3. Vitamin K2

Vitamin K2 (menaquinone) is considered to assist the gamma-carboxylation of osteocalcin, which is produced by osteoblasts during bone matrix formation. It has been reported that a high serum level of undercarboxylated osteocalcin is a risk factor for fractures in elderly women, which could be used as an independent marker of fracture. Administration of menatetrenone leads to decreased serum levels of undercarboxylated osteocalcin [149,150,151].

### 4.6. Novel Targets, Novel Approach, and Experimental Materials for the Prevention of Osteoporosis

#### 4.6.1. Combination Therapies

Combination therapies with denosumab and estrogen-like drugs (e.g., raloxifene) exhibited superior efficacy over monotherapy with an additional 36% reduction in fracture risk [152], and a faster BMD increase and total hip BMD recovery [153]; however, protection from fracture is not known. On the other hand, the combination of BPs and teriparatide does not produce any meaningful benefit over monotherapy [154]. Instead, the sequential usage of these drugs for accomplishing long-term management of osteoporosis risk is encouraged. For instance, estrogen and raloxifene can be subscribed to relatively young postmenopausal women, followed by teriparatide and abaloparatide, which are appropriate for patients at imminent risk of vertebral fracture. Moreover, with those therapies, continuation of treatment with BPs or denosumab should be considered in patients with a high risk of fracture.

#### 4.6.2. Experimental Findings for Novel Targets

It has been suggested that pharmacological modulation of peripheral cannabinoid receptor types (CB1 and CB2) regulates bone mass [155]. Inactivation of CB1 by its antagonist, AM251, or stimulation of TRPV1 channel by its selective agonist, resiniferatoxin, reduced calcium deposition by osteoblasts through downregulation of Runx2, OPG, alkaline phosphatase [156]. Conversely, CB2 stimulation by its selective agonist, JWH-133, increased bone calcium deposition through the inhibition of RANKL secretion [157]. Another CB2 selective agonist, HU308, also caused the inhibition of RANKL-induced osteoclast formation in RAW 264.7 culture in vitro [158].

p38 MAPK is known to be an essential regulator of RANKL-mediated osteoclastogenesis and bone loss [159]. TAT-TN13, a 13-amino-acid of thioredoxin-interacting protein, inhibiting p38α, significantly suppressed RANKL-mediated differentiation of RAW 264.7 cells and bone marrow macrophages into mature osteoclasts [160,161]. Another strategy for reducing the effect of RANKL is to modulate its receptor. For instance, LGR4 is a RANKL receptor that negatively regulates osteoclast differentiation and bone resorption [162].

Understanding the detailed mechanisms underlying osteoclastogenesis and osteoblastogenesis has led to the discovery of various materials, especially natural products that target either process. For example, petunidin, a B-ring 5′-O-methylated derivative of delphinidin promotes osteoblastogenesis [163], whereas kukoamine B, which is fractionated from Lycii Radicis Cortex (LRC) extract, enhances osteoblast differentiation [164].

#### 4.6.3. Stem Cells

Stem cell-based therapies are becoming increasingly important in the treatment of chronic and long-lasting diseases, including osteoporosis, as they could enable curative and personalized regenerative medicine approaches. Several different types of stem cells have been evaluated to modulate osteoporosis, including embryonic, induced pluripotent, and MSCs [165,166,167,168,169]. Among them, MSCs are critical candidates for bone regenerative medicine, as they have advantages over other types of stem cells clinically, including ease of harvesting, immunosuppressive outcomes, and fewer ethical concerns [170,171,172]. It has also been found that bioactive molecules, such as IGF-1, TGF-β, vascular endothelial growth factor (VEGF), hepatocyte growth factor (HGF), angiogenin, and IL-6 that are derived from MSCs, can support bone regeneration to a great extent [173,174,175,176]. Furthermore, exosomes released by MSCs have been demonstrated to have a promising effect on bone remodeling and the prevention of bone loss in vivo [177,178].

MSC transplantation is feasible and its effects on bone formation have been previously described [179]. MSCs can directly repair the pathological area and also differentiate into osteoblasts, which is their endogenous role in bone formation. Bone marrow-derived MSCs with high osteogenic differentiation potential are considered to be a reliable and effective source for osteoporosis MSC therapies [180,181,182,183]. MSC transplantation has been conducted in osteoporotic animal models and humans, and it could indeed support microenvironment, promote bone regeneration and have paracrine effects. Thus, the secretome and/or exosomes from MSCs transplantation might be key to regulating osteoporosis, in addition to the cell therapy itself [184]. In line with this, there are over 1000 clinical trials of MSC transplantation therapies registered with ClinicalTrials.gov (http://www.Clinicaltrials.gov/). Many cases involve bone diseases and conditions, such as osteoarthritis, osteogenesis imperfecta, rheumatoid arthritis, and osteoporosis. Research efforts will be made in the future to clarify the clinical application of MSCs in osteoporosis [185].

#### 4.6.4. siRNAs and Other Inhibitors

Efforts have been made to achieve specific target gene downregulation using siRNAs. As a result, many gene targets in osteoporosis, such as *RANKL/RANK*, *DKK-1*, *SOST* can be manipulated by using siRNAs instead of chemical-based inhibitors. However, this strategy requires further investigation for clinical relevance [186]. Similarly, apart from antibodies, aptamers can be investigated for targeting RANKL [187,188]. Aptamers are small single-stranded oligonucleotides capable of binding target molecules with high affinity and specificity. They are easier and cheaper to generate, not immunogenic, and thus, many ongoing clinical trials are investigating their potential use [189].

The molecular mechanisms of osteoporosis and therapeutic agents are summarized in Figure 2.

## 5. Conclusions

Osteoporosis is an increasingly prevalent condition as the aging population grows fast globally. It causes more than 8.9 million fractures per year worldwide [190]. Not only in western countries, but also in East Asian countries, such as China, Korea, and Japan, many elderly women and men already have increased osteoporotic fracture risks. Osteoporotic fractures may lead to significant functional limitations and increased mortality. Thus, a timely diagnosis, prescription of medication, as well as the management of this disease are important. Although the optimal period of pharmacological treatment and the starting age is controversial, novel strategies and emerging therapies could provide more options for patients and clinicians. To date, BPs remain the first-line and most cost-effective medicine for osteoporosis, but there is also concern about their long-term use due to safety issues. As mentioned above, the first goal of osteoporosis treatment using pharmaceuticals is to reduce the risk for fracture. To achieve this, adequate investigations should be conducted and a proper diagnosis should be given. Since BMD alone cannot predict the risk for fracture, new cutting-edge technologies, including next-generation sequencing, genome-wide screening and assessment, and stem cell therapeutics should be considered. In conclusion, by gaining a better understanding of the molecular mechanisms underlying osteoporosis, enabling and using new emerging technologies, it is possible to achieve better outcomes in patients with osteoporosis.

## Figures and Tables

**Figure 1 ijms-21-07623-f001:**
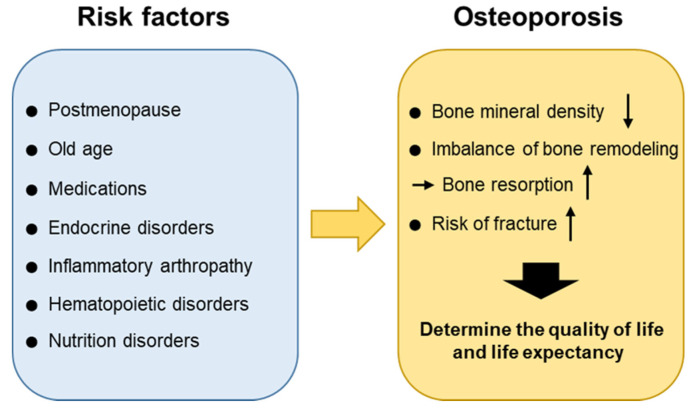
Risk factors for osteoporosis. Risk factors can cause an imbalance in the remodeling process leading to osteoporosis. The postmenopausal period and old age are the major causes of osteoporosis.

**Figure 2 ijms-21-07623-f002:**
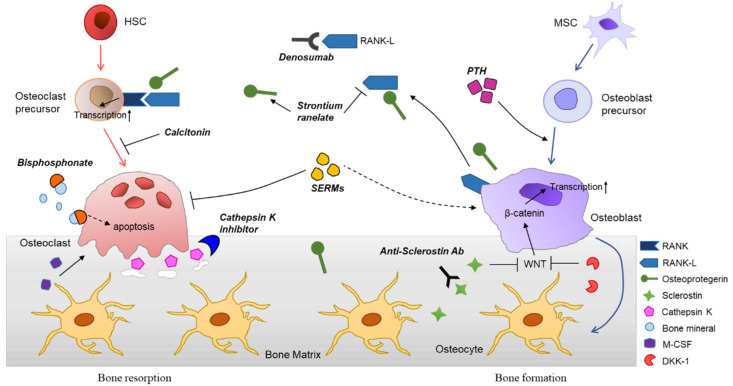
Differentiation of bone cells, bone remodeling process, and various therapeutic agents for osteoporosis. Hematopoietic stem cells (HSCs) are differentiated into osteoclasts, mediated through stimulation of receptor activator of nuclear factor kappaB ligand (RANKL), generated from osteoblasts. Osteoclasts can be further maturated by monocyte-colony stimulating factor (M-CSF). The bone resorption occurs by matrix metalloproteinases and cathepsin K, secreted by mature osteoclasts. Osteoblasts are derived from the mesenchymal stem cells (MSCs) and involved in bone formation. The major roles of osteoblasts in bone remodeling are activation of osteoclasts differentiation and generation of bone cells including osteocytes. Various signaling molecules, such as insulin-like growth factor 1 (IGF-1), transforming growth factor β (TGF-β), and Wnt induce osteoblast differentiation. Osteocytes are embedded in the bone matrix and orchestrate the bone remodeling. They promote bone formation by releasing osteoprotegerin together with osteoblasts, whereas suppress osteoblastogenesis by secretion of sclerostin and Dickkopf-related protein 1 (DKK-1), inhibitors of Wnt signaling. Many therapeutic agents are being developed based on the molecular biology of bone and used clinically. Anti-resorptive agents are bisphosphonates (BPs), anti-RANKL antibodies (e.g., denosumab), selective estrogen receptor modulators (SERMs), and calcitonin. Parathyroid hormone (PTH) analogues, strontium ranelate, and anti-sclerostin antibodies can be categorized as anabolic agents for osteoporosis.

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
