# Peer review of "Molecular Mechanisms and Emerging Therapeutics for Osteoporosis"

_ijms, 2020, doi:10.3390/ijms21207623_

Round 1

Reviewer 1 Report

In this review article, Noh et al. performed a literature search and combined with their recent publications on the molecular mechanisms regulating hematopoietic stem cells, osteoclastogenesis, MSCs and osteoblastogenesis discuss the potential therapeutic approaches for osteoporosis. This review is well written and contain most of the recent developments in the area of osteoporosis research. However, the advances in the knowledge about the role of bone marrow adipocytes involved in pathogenesis of osteoporosis and/or bone loss disorders are skipped. This reviewer suggests that the impact of bone marrow adipocytes, their cross-talk with osteoblasts and osteoclasts and future therapeutic perspectives should be discussed in the appropriate sections to increase the review quality as well as up-to-date information included in the review.

  1. The role of bone marrow adipocytes in “Molecular mechanism of bone remodeling and osteoporosissection:discuss role of adipogenic transcription factors as well as the adipogenic molecules in normal and osteoporotic conditions.

  1. PPARg: The master transcription factor in Molecular mechanism section: discuss role of PPARg, and other transcription factors in the intertwining of adipogenesis and osteoclastogenesis, and the suppression of intracellular osteoblastogenic pathways by adipogenic transcription factors and signaling pathways.

  1. Discuss the influence of novel therapeutics on adipocyte differentiation program under the section “Therapeutic approach and novel strategies”

Line 146            3.1.1. Oteoclasts spelling error, to be correctly read as Osteoclasts

Author Response

In this review article, Noh et al. performed a literature search and combined with their recent publications on the molecular mechanisms regulating hematopoietic stem cells, osteoclastogenesis, MSCs and osteoblastogenesis discuss the potential therapeutic approaches for osteoporosis. This review is well written and contain most of the recent developments in the area of osteoporosis research. However, the advances in the knowledge about the role of bone marrow adipocytes involved in pathogenesis of osteoporosis and/or bone loss disorders are skipped. This reviewer suggests that the impact of bone marrow adipocytes, their cross-talk with osteoblasts and osteoclasts and future therapeutic perspectives should be discussed in the appropriate sections to increase the review quality as well as up-to-date information included in the review.

1. The role of bone marrow adipocytes in “Molecular mechanism of bone remodeling and osteoporosis“ section: discuss role of adipogenic transcription factors as well as the adipogenic molecules in normal and osteoporotic conditions.

Response 1:

We appreciate the reviewer’s comment. Bone marrow adipose tissue have achieved attractions because of their multiple roles in bone remodeling. As the reviewer suggested, we summarized this issue and added several references. We provide the relevant changes made to the manuscript, in color blue.

In page 10, “Molecular mechanism of bone remodeling and osteoporosis” section,

3.1.4. Adipocyte

Bone marrow adipogenesis plays a critical role in bone loss associated with aging as well as diseases such as diabetes mellitus. The differentiation of adipocytes derived from MSC in bone marrow competes with osteoblastogenesis by its nature of lineage allocation while the mature adipocytes express RANKL and promote osteoclastogenesis [83,84]. Notably, among the adipogenic transcription factors, including CCAAT/enhancer binding protein alpha (C/EBPa), C/EBPb, and peroxisome proliferator-associated receptor gamma (PPARg), PPARg is considered as a master factor for osteoclast differentiation from HSCs [85]. PPARg downregulates Wnt/β-catenin signaling pathway in both MSC and HSC, resulting in inhibition of osteoblastogenesis and activation of osteoclast function, respectively [86]. In osteoclast, PPARg-mediated expression of peroxisome proliferator-activated receptor-gamma coactivator-1 beta (PGC1b) promotes mitochondria biogenesis, which in turn increases bone resorption [87].

In addition to the regulation by transcription factors in bone cells and adipocytes, paracrine effects by adipokines such as chemerin, resistin, visfatin, leptin, adiponectin, and omentin-1 secreted from adipocytes have shown to involve in bone remodeling. Interestingly, a cognate receptor for chemerin, chemokine-like receptor 1 (CMKLR1) was identified in MSCs, and chemerin/CMKLR1 signaling suppresses Wnt/β-catenin and Notch signaling pathways, resulting in inhibition of osteoblastogenesis [86]. Furthermore, chemerin/CMKLR1 signaling induces expression of NFATc1 in HSCs, a key transcription factor for osteoclast differentiation, demonstrating that chemerin may potentiate bone loss in disease states [88]. Other adipokines such as visfatin and resistin represented similar activity, although the role of visfatin in bone remodeling is controversial [83]. Collectively, bone formation is regulated by the cell-to-cell contact involving RANK-RANKL in normal condition, however adipocytes as well as adipokines can play critical roles in bone-loss associated diseases.

2. PPARg: The master transcription factor in Molecular mechanism section: discuss role of PPARg, and other transcription factors in the intertwining of adipogenesis and osteoclastogenesis, and the suppression of intracellular osteoblastogenic pathways by adipogenic transcription factors and signaling pathways.

Response 2:

We appreciate the reviewer’s comment and we described the molecular mechanisms of transcription factors in the same section on 3.1.4. in page 10. Please see the response 1.

3. Discuss the influence of novel therapeutics on adipocyte differentiation program under the section “Therapeutic approach and novel strategies”

Response 3:

Adipocyte differentiation program and/or some of adipokines are novel therapeutic targets for treating osteoporosis. Importantly, the therapeutic efficacy can be potentiated due to the dual effects on both inhibition of osteoclastogenesis and activation of osteoblastogenesis. Thus, we summarized several novel findings including investigational approaches in the “4.4. Emerging therapies and investigational agents for targeting bone formation” section.

The manuscript has been greatly improved and we really appreciate the reviewer’s comment.

In page 17, “Therapeutic approach and novel strategies” section,

4.4.3. Suppression of bone marrow adipogenesis and/or adipokines (chemerin)

Approaches to suppress bone marrow adipogenesis and/or chemerin may have therapeutic value for treating osteoporosis due to the advantages resulting from inhibition of bone resorption and enhanced bone formation [86]. In this context, PPARg is a good target because of its dual role in MSC-derived adipogenesis as well as HSC-derived osteoclastogenesis. PPARg can be directly targeted by its antagonist, GW9962 [139], or by regulation of upstream signaling molecules, for instance, lipid mediators. Recently, it has been reported that upregulation of sphingosine-1-phosphate (S1P) by inhibition of S1P lyase showed therapeutic benefit for treating osteoporosis, mainly mediated through PPARg suppression and enhanced bone formation [140]. Similarly, the level of glucosylceramide (GlcCer) is also important for PPARg activity, and inhibition of GlcCer synthase has been shown to suppress adipogenesis [141].

Therapeutic effect on osteoporosis can be also achieved by targeting adipokines such as chemerin and visfatin [86]. Since chemerin/CMKLR1 signaling inhibits Wnt/β-catenin and Notch signaling pathway, bone-anabolic activity can be reversed by suppression of chemerin. Interestingly, this suppression of adipogenesis was found to be restricted to the bone marrow adipose tissue. Furthermore, osteoporotic patients treated with bone anabolic agents (e.g. teriparatide) has shown less adipose tissue in bone marrow, indicating that osteoporotic therapies targeting the osteoblast/adipocyte/osteoclast complex are promising [142].

4. Line 146 3.1.1. Oteoclasts spelling error, to be correctly read as Osteoclasts

Response 4:

We corrected the typo.

Reviewer 2 Report

Well-written review.

Major Changes:

Include a few paragraphs about drugs that help increase osteoblasts activity and numbers (e.g. Teriparatide)

Donepezil is another drug involved in the treatment of osteoporosis (can cite this paper PMID: 32407731)

Minor Changes:

In 4.3.2, given an explanation of why strontium is restricted to only severe osteoporosis treatment

Line 472, microCT is another non-invasive way to look at bone health

Author Response

Well-written review.

Major Changes:

1. Include a few paragraphs about drugs that help increase osteoblasts activity and numbers (e.g. Teriparatide)

Response 1:

We have sections on “4.3. Anabolic agents” as well as “4.4. Emerging therapies and investigational agents for targeting bone formation” that introduce several drugs for increasing osteoblast differentiation including Teriparatide (page 14, section 4.3.1.). However, we skipped the important findings and recent updates on bone marrow adipose tissue and adipokines in the previous manuscript, so we added several therapeutic approaches that can be considered as anabolic agents.

We provide the relevant changes made to the manuscript, in color blue.

We really appreciate the reviewer’s comment.

In page 17, “Therapeutic approach and novel strategies” section,

4.4.3. Suppression of bone marrow adipogenesis and/or adipokines (chemerin)

Approaches to suppress bone marrow adipogenesis and/or chemerin may have therapeutic value for treating osteoporosis due to the advantages resulting from inhibition of bone resorption and enhanced bone formation [86]. In this context, PPARg is a good target because of its dual role in MSC-derived adipogenesis as well as HSC-derived osteoclastogenesis. PPARg can be directly targeted by its antagonist, GW9962 [139], or by regulation of upstream signaling molecules, for instance, lipid mediators. Recently, it has been reported that upregulation of sphingosine-1-phosphate (S1P) by inhibition of S1P lyase showed therapeutic benefit for treating osteoporosis, mainly mediated through PPARg suppression and enhanced bone formation [140]. Similarly, the level of glucosylceramide (GlcCer) is also important for PPARg activity, and inhibition of GlcCer synthase has been shown to suppress adipogenesis [141].

Therapeutic effect on osteoporosis can be also achieved by targeting adipokines such as chemerin and visfatin [86]. Since chemerin/CMKLR1 signaling inhibits Wnt/β-catenin and Notch signaling pathway, bone-anabolic activity can be reversed by suppression of chemerin. Interestingly, this suppression of adipogenesis was found to be restricted to the bone marrow adipose tissue. Furthermore, osteoporotic patients treated with bone anabolic agents (e.g. teriparatide) has shown less adipose tissue in bone marrow, indicating that osteoporotic therapies targeting the osteoblast/adipocyte/osteoclast complex are promising [142].

2. Donepezil is another drug involved in the treatment of osteoporosis (can cite this paper PMID: 32407731)

Response 2:

In line with the response 1, we included the description on donepezil in the section about anabolic agents, as donepezil has been considered to promote osteoblastogenesis.

The manuscript has been greatly improved. And we really appreciate the reviewer’s comment. We also added a few new references including PMID:32407731 as the reviewer suggested.

In page 16, “Therapeutic approach and novel strategies” section,

4.4.2. Donepezil

Donepezil is a medication that have been widely used in the treatment of Alzheimer’s disease and other dementias since the mid-1990s [135]. It is a reversible acetylcholinesterase (AChE) inhibitor. Notably, acetylcholine receptors (AChR) are expressed in bone cells, of which stimulation by inhibiting the action of AChE and increasing the amount of Ach can have anabolic effects in bone. In line with it, AChE is highly expressed on bone cells, especially during osteoblastogenesis [136], suggesting that AChE can be a therapeutic target for treating osteoporosis. Interestingly, donepezil showed beneficial effects on bone turnover, associated with reduction of hip fracture risk in Alzheimer’s disease patients [137,138].

3. In 4.3.2, given an explanation of why strontium is restricted to only severe osteoporosis treatment

Reponse 3:

We agree with the reviewer’s point. We added a sentence to explain the reason.

In page 15, 4.3.2. Strontium ranelate,

Thus, it increases bone formation and decreases bone resorption; however, due to the increased risk of heart problems, the use of strontium is restricted to severe osteoporosis where other treatments are not available.

4. Line 472, microCT is another non-invasive way to look at bone health

Response 4:

We really apologize, but we could not find the relevant content in the previous manuscript. However, as the reviewer commented, we thought that it could be more helpful to include the information about the diagnosis method to check bone health. Therefore, we included a sentence.

In page 11, 4. Therapeutic approach and novel strategies,

To assess drug efficacy, non-invasive methods such as dual-energy vertebral assessment program (DXA) or micro-computed tomographic (microCT) can be used.
